# Classification of Codling Moth-Infested Apples Using Sensor Data Fusion of Acoustic and Hyperspectral Features Coupled with Machine Learning

Nader Ekramirad [1], Alfadhl Y. Khaled [1], Kevin D. Donohue [2], Raul T. Villanueva [3] and Akinbode A. Adedeji [1,*]

1   Department of Biosystems and Agricultural Engineering, University of Kentucky, Lexington, KY 40546, USA
2   Department of Electrical and Computer Engineering, University of Kentucky, Lexington, KY 40506, USA
3   Department of Entomology, University of Kentucky, Princeton, KY 42445, USA
*   Correspondence: akinbode.adedeji@uky.edu; Tel.: +1-(859)-218-4355

**Abstract:** Codling moth (CM) is a major apple pest. Current manual method of detection is not very effective. The development of nondestructive monitoring and detection methods has the potential to reduce postharvest losses from CM infestation. Previous work from our group demonstrated the effectiveness of hyperspectral imaging (HSI) and acoustic methods as suitable techniques for nondestructive CM infestation detection and classification in apples. However, both have limitations that can be addressed by the strengths of the other. For example, acoustic methods are incapable of detecting external CM symptoms but can determine internal pest activities and morphological damage, whereas HSI is only capable of detecting the changes and damage to apple surfaces and up to a few mm inward; it cannot detect live CM activity in apples. This study investigated the possibility of sensor data fusion from HSI and acoustic signals to improve the detection of CM infestation in apples. The time and frequency domain acoustic features were combined with the spectral features obtained from the HSI, and various classification models were applied. The results showed that sensor data fusion using selected combined features (mid-level) from the sensor data and three apple varieties gave a high classification rate in terms of performance and reduced the model complexity with an accuracy up to 94% using the AdaBoost classifier, when only six acoustic and six HSI features were applied. This result affirms that the sensor fusion technique can improve CM infestation detection in pome fruits such as apples.

**Keywords:** apples (*Malus domestica*); codling moth; sensor fusion; hyperspectral image; acoustic; machine learning





## 1. Introduction

Apples are one of the most valuable fruits in the USA with domestic consumption and total exports of around 4.1 and 0.87 million metric tons, respectively [1]. However, the codling moth (CM) pest causes significant damage to apples pre- and post-harvest. The presence of a CM larva can cause the rejection of fruit shipments from most U.S. export destinations [2] and up to a 59% reduction in value when infested apples are diverted to other low-value uses [3]. To improve the detection approach, there is a need to develop rapid, effective, and accurate nondestructive detection methods for CM-infested apples [4–6].

Generally, fruits have complex and dynamic textures with different characteristics [7]. As a result, only limited information of fruit samples can be obtained using an individual sensing technique [8]. Thus, merging data from different sensors can provide comprehensive information about the characteristics of fruits and improve the prediction and classification rates through a better understanding of the internal and external states of the produce. Information fusion strategies have been defined as methods of fusing data from

different sensors or knowledge from different models, while the relationship between the fused information and the target parameter is represented as a mathematical model [9,10]. Three levels of fusion strategies have been defined based on the type of information to be fused: (1) measurement or low-level fusion, (2) feature or mid-level fusion, and (3) decision or high-level fusion [11]. In the first level of fusion, the raw data from the sensors are integrated into a new dataset for further processing. This strategy suffers from high amounts of redundant and noisy data [12]. For the second level of fusion, the extracted features from each sensing technique are fused as inputs to the final model. This method can address the redundancy and noise issues to achieve improved results [13]. In the third level of fusion, the outputs of multiple models are combined for a full evaluation of the final decision. For example, the majority voting method takes into account the results of many classifiers to provide an overall decision. While the decision fusion strategy potentially reduces the interference by the limitations of different models, it has the risk of losing important information in the raw data [11].

Recently, fusion strategies have been used in studies on defect detection and quality assessment of fruits. Liu et al. [14] applied a mid-level/feature fusion method based on HSI and Electrical nose (E-nose) data for fungal contamination detection in strawberries. They concluded that while the raw data fusion of HSI and E-nose resulted in a low prediction rate and high processing time, the feature fusion method improved the detection accuracy compared with each of the individual sensing methods. In another study, the application of fusion of HSI and olfactory sensors for tea quality evaluation was investigated [15]. From the results presented, the accuracy of the models for evaluating tea improved from 75% for the individual sensor data to 92% when applying the fused data.

Codling moth pest attacks lead to damage to both the external and internal physico-chemical characteristics of apples [16]. While the fusion of different sensing methods can provide comprehensive and combined information related to the infestation, individual sensing techniques will only capture one (or a few) of the many aspects of infestation damage. For example, HSI provides physical and chemical information from the top layers of fruit tissue and flesh [14], however, it is not able to capture data from the core of apples. On the other hand, vibrational/acoustic methods can be used to monitor and detect infested apples through sensing, either by the activities of the insects that bore deep into the fruit or the internal textural changes related to infestation [5,17]. The outputs of the two sensing systems can be fused and analyzed using multivariate approaches to improve the pattern recognition results for classifying infested apples. Because the capability of rapid detection by HSI and acoustic can be negatively affected by large data dimensionality, the specific objective was to perform mid-level fusion with feature extraction and selection from the raw HSI and acoustic data and then develop the fusion models based on the multiple optimum features. Thus, in this study, we investigated the application of the sensor data fusion approach (HSI and acoustic) for improving classification accuracy in the detection of postharvest CM infestation in apples.

## 2. Materials and Methods

### 2.1. Sample Preparation

The apple samples used in the experiments were organic Gala, Fuji, and Granny Smith cultivars purchased from a commercial market in Princeton, KY, USA in October 2020. After careful inspection, 60 apple samples without any form of mechanical damage that were similar in size, diameter, and shape were chosen from each cultivar (180 samples in total). The apples were then disinfected against fungal and bacterial decay in a 0.5% ($v/v$) sodium hypochlorite solution according to Louzeiro et al. [18]. The samples were washed with distilled water and dried in the open air at ambient conditions at 25 ± 2 °C in the Lab (Department of Entomology, University of Kentucky, Princeton, KY, USA). To artificially infest the apples, a first instar CM larva was placed near the calyx end of each apple in an isolated cup (8 cm bottom diameter, 10 cm top diameter, and 10 cm high) with a plastic lid for respiration purposes. Figure 1 shows an example of the external and

internal views of a CM-infested apples. The apples of each cultivar were divided into 20 control and 40 infested groups and stored in an environmental control chamber at 27 °C and 85% relative humidity for three weeks to cause infestation to occur. Hyperspectral data acquisition was carried out in the Food Engineering lab at Biosystems and Agricultural Engineering Department, University of Kentucky, Lexington, KY, USA.

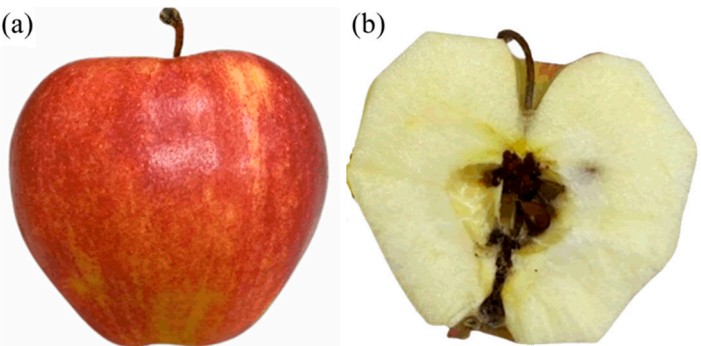

**Figure 1.** A typical Codling moth (CM) infested apple. (**a**) External view; (**b**) internal view.

## 2.2. Hyperspectral Image Acquisition and Spectral Extraction

The short wave near-infrared (SWNIR) HSI system in the spectral range of 900–1700 nm was used to acquire hyperspectral images of healthy and infected apples for each cultivar (Figure 2). This system was formed using an imaging spectrograph (N17E, Specim, Oulu, Finland), an InGaAs camera (Goldeye infrared camera: G-032, Allied Vision, Stradtroda, Germany), a stepping-motor-driven moving stage (MRC-999–031, Middleton Spectral Vision, Middleton, WI, USA), and a 150 W halogen lamp (A20800, Schott, Southbridge, MA, USA). The hyperspectral imaging system is a pushbroom (line scanning) type. To acquire clear images, the parameters of the sample stage speed, exposure time of the camera, halogen lamp angle, and vertical distance between the lens and the sample, were set to 10 mm/s, 40 ms, 45°, and 25 cm, respectively. The samples were placed on the sample stage and captured in a line scanning or pushbroom mode. The acquired hyperspectral images contained wavelength bands in "*.raw" format along with a header file in "*.hdr" format. Three scans were acquired for each apple sample in the stem, calyx, and side-view orientations.

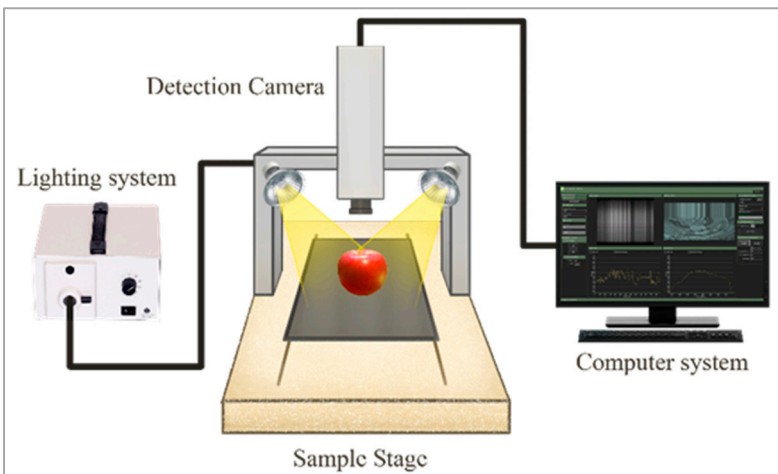

**Figure 2.** A schematic of the hyperspectral imaging (HSI) system [4].

## 2.3. Acoustic Impulse Response Test and Signal Recording

After hyperspectral image acquisition, each sample was used for the acoustic test. A schematic of the acoustic impulse response test is shown in Figure 3. It consists of two main parts: the acoustic recording unit and the impulse or knocking unit. The unit used

for conducting the impulse or knocking test comprised of two primary parts, namely an impulse generator and a mechanical support system. This arrangement was intended to make the apple more secure when it was attached to other parts with respect to each other, as illustrated in Figure 3. The support system was fabricated using standard lab metalware and mounted on an individual ring stand with a cast-iron base to minimize any resonance effects (American Educational 7-G15-A).

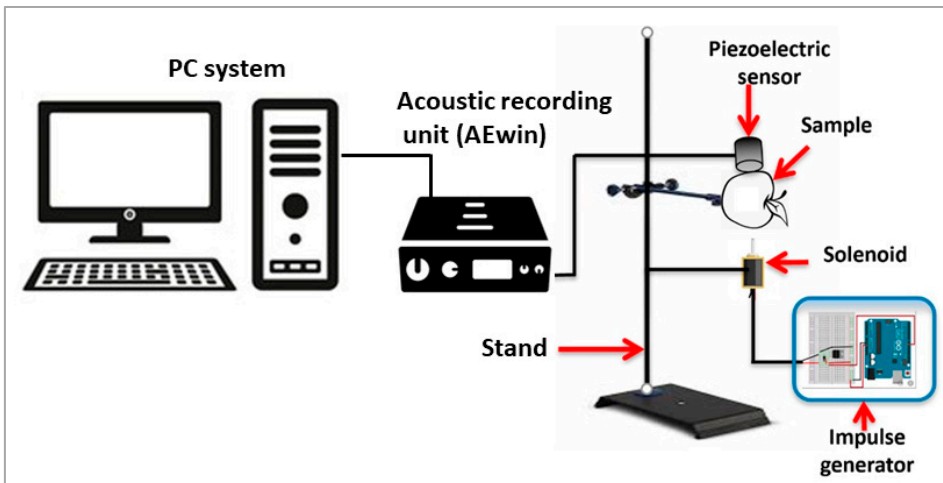

**Figure 3.** A schematic diagram of the acoustic impulse response system for data acquisition from an apple.

The apple was carefully positioned within a three-prong gripper, secured with an actuating screw. The other grippers were adjusted vertically and laterally. A spacer attached to the end of the solenoid ensured a consistent distance from the apple's surface. The flexible setup accommodated different sizes and shapes of apples while ensuring firm and consistent testing. The experiment used a precise solenoid impulse generator controlled by a microcontroller. The solenoid model chosen had a nose with a 6.35 mm radius on the armature to deliver the impact. A push button triggered the impulse, connected to the microcontroller, configured to generate a 50 µs output pulse with a hold-off. The duration was sufficient to ensure the solenoid reached maximum extension at 9V. The pulse was transmitted through a resistor to a TIP-31c NPN transistor to handle the current and EMF kick of the solenoid. Power was supplied using a 9V DC adapter.

The acoustic recording unit was a custom-designed system to record the high-frequency acoustic response signals from apples generated by the impulse/knocking test. This system consisted of a contact piezoelectric sensor (R6$\alpha$-SNAD 52, Physical Acoustics Corporation, West Windsor Township, NJ, USA) with a frequency range of 35 to 100 kHz, a preamplifier (model1220A, Physical Acoustics Corp., West Windsor Township, NJ, USA), an I/O board (PCI-2, Physical Acoustics Corp., West Windsor Township, NJ, USA), and signal processing software (AEwin by MISTRAS).

To reduce the ambient noise, the acoustic impulse response experimental unit was set above an isolated table that had a 15 cm layer of sand, topped with a 5 cm slab of granite with acoustic padding. This unit was in a room with a concrete padded floor built on 20 cm of gravel above the loam soil bed in an isolated room in the Food Engineering Lab at the Biosystems and Agricultural Engineering Department, University of Kentucky, Lexington, KY, USA. To carry out each test, an apple was placed between the sensor and the impulse generator (solenoid). The signal recording for each test was performed for 10 s with two impulses for each apple, where the first impulse was generated in the fifth second and the second impulse in the tenth second. The acoustic signals derived from the knocking impulse on apples were collected and processed by different signal processing methods, and then the time-domain and frequency-domain features of the vibration acoustic signals were extracted for use in the machine learning classification models.

After manually segmenting the actual impulse moment from the entire signal, 21 important time and frequency domain features were extracted (Table 1) using a code created in MATLAB (Release 2020b, The MathWorks, Inc., Natick, MA, USA). With these features as the variables (columns) for all samples (as rows), the dataset was built for use in machine learning classification. Moreover, these features were concatenated with the HSI features to build the data fusion models.

**Table 1.** Selected time- and frequency-domain features.

| No. | Feature Name | Domain Type | Explanation |
|-----|-------------|-------------|-------------|
| 1 | Average signal level | Time | A technique in signal processing is used to enhance the signal's strength in comparison to the noise that is interfering with it. |
| 2 | Variance | Time | The mathematical expression represents the average value of the squared difference between a random variable and its mean. |
| 3 | Kurtosis | Time | A numerical indication of the degree of dispersion or flattening of the probability distribution of a random variable that takes real values. |
| 4 | Skewness | Time | The presence of skewness or distortion in a normal distribution or bell curve, which is symmetric by definition, within a given dataset. |
| 5 | Mean absolute deviation | Time | The mean of the absolute deviations of each data point from the arithmetic mean, which measures the average distance between the data points and the mean. |
| 6 | Root mean square | Time | The square root of the mean square. |
| 7 | Entropy | Time | A metric used to evaluate the distribution of power across the spectral range of a signal. |
| 8 | Mean rise time | Time | The average duration required for a signal to transition from a predetermined low value to a predetermined high value. |
| 9 | Absolute energy | Time | The calculation obtained by adding the squares of individual signal values. |
| 10 | Area under curve | Time | The summation signal values. |
| 11 | Signal strength | Time | A measure of the power of the signal. |
| 12 | Average value of peaks | Time | The mean value of the peak amplitudes crossing the threshold. |
| 13 | Number zero crossing | Time | The momentary position where there is an absence of frequency components. |
| 14 | Number of peaks | Time | The number of maximum amplitudes. |
| 15 | Energy spectral density | Frequency | The distribution of the energy of the signal in the frequency domain. |
| 16 | Maximum power spectral density | Frequency | The peak power level exhibited by a signal over the range of frequencies it contains. |
| 17 | Centroid | Frequency | The average location of all points within a signal is calculated through the arithmetic mean. |
| 18 | Peak frequency | Frequency | The frequency of maximum power. |
| 19 | Power bandwidth | Frequency | The variation in power between the highest and lowest frequencies within a continuous frequency range. |
| 20 | Maximum spectral entropy | Frequency | A technique used for estimating spectral density. |
| 21 | Fast Fourier transform mean coefficient | Frequency | The average of the values of a signal in the frequency domain. |

*2.4. Data Fusion Strategies*

Data fusion is defined as the fusion of the data acquired using different sensors [19]. In this study, low-level and mid-level data fusion strategies were implemented to combine information from hyperspectral and acoustic datasets for CM-infestation detection in apples. In the low-level fusion, the raw hyperspectral and acoustic datasets were concatenated into a single matrix by merging them along the rows. This resulted in a combined data matrix with the same number of rows as the number of samples. The columns represent the combined variables from each dataset (241 spectral and 21 acoustic). However, because the features from different sensors had different scales, a z-score normalization was used for rescaling purposes before building the model. In mid-level fusion, the extracted features from the hyperspectral dataset using the PCA method were fused with the optimum acoustic features selected by the Pearson correlation method (six HSI and six acoustic

features). The merged data matrices from the low-level and mid-level methods were then used to build multivariate calibration models.

Principal component analysis (PCA) has been widely applied for dimensionality reduction in large feature datasets usually obtained from the HSI method to reduce the possibility of overfitting [20–22]. PCA is a linear method that transforms features by axis rotation to align the first principal component with the direction of maximum variance. The other principal components (PCs) are perpendicular to the previous components and are represented as linear combinations of the variables. Using only a few of the first PCs, it is possible to represent a significant amount of the total variance of the entire dataset [23].

*2.5. Classification Models*

After creating the datasets, to build and compare the different classifiers, the PyCaret (Version 2.3.10) machine learning library in Python was used. Different classification algorithms were used for the sorting process, including support vector machine (SVM), random forest (RF), k-nearest neighbors (kNN), decision trees (DT), linear discriminant analysis (LDA), Naïve Bayes (NB), Ridge, gradient boosting (GB), quadratic discriminant analysis (QDA), extra trees (ET), and AdaBoost (AB), to build the retrieval. Several studies have used these models in various classification applications [20–27].

The results of these models were analyzed and compared, and the best model obtained was the ensemble AdaBoost method based on the total accuracy, recall, precision, and F1 score. Then the average values for the accuracy, recall, precision, and F1-score were calculated in a fivefold cross-validation process. These values were calculated as follows:

$$\text{Accuracy} = \frac{\text{TP} + \text{TN}}{\text{TP} + \text{FP} + \text{TN} + \text{FN}} \tag{1}$$

$$\text{Recall} = \frac{\text{TP}}{\text{TP} + \text{FN}} \tag{2}$$

$$\text{Precision} = \frac{\text{TP}}{\text{TP} + \text{FP}} \tag{3}$$

$$\text{F1 score} = 2 \ \times \ \frac{\text{Recall} \times \text{Precision}}{\text{Recall} + \text{Precision}} \tag{4}$$

where true positive (TP) and true negative (TN) are the correctly predicted samples that belong to their actual class. False positive (FP) and false negative (FN) values are obtained, when the predicted level conflicts with the actual level. Precision, which is the positive predictive value, indicates the number of samples correctly classified as infested. On the other hand, recall, which is the true positive rate, is related to the number of samples that belong to the infested group and were predicted to be positive, including those that were incorrectly classified as healthy by the model.

## 3. Results and Discussions

*3.1. Feature Dimensionality of HIS Data Based on PCA*

In this study, the PCA was used to reduce the dimensions of the preprocessed spectra from 241 to 10 and even 6 features before building the classification models. Based on these results, the accumulated variance represented by only the first three PCs for all the three apple cultivars was more than 99% of the total variance in each case (Figure 4). Therefore, it was expected that the samples would be classified using this limited number of PCs as the inputs to machine learning models for the classification of apples. Similar results were reported by Moscetti et al. [28] for the application of PCA on the NIR spectroscopy data of non-infested and infested olive fruits with the first two PCs accounting for 98.3% of the total variance in the spectra.

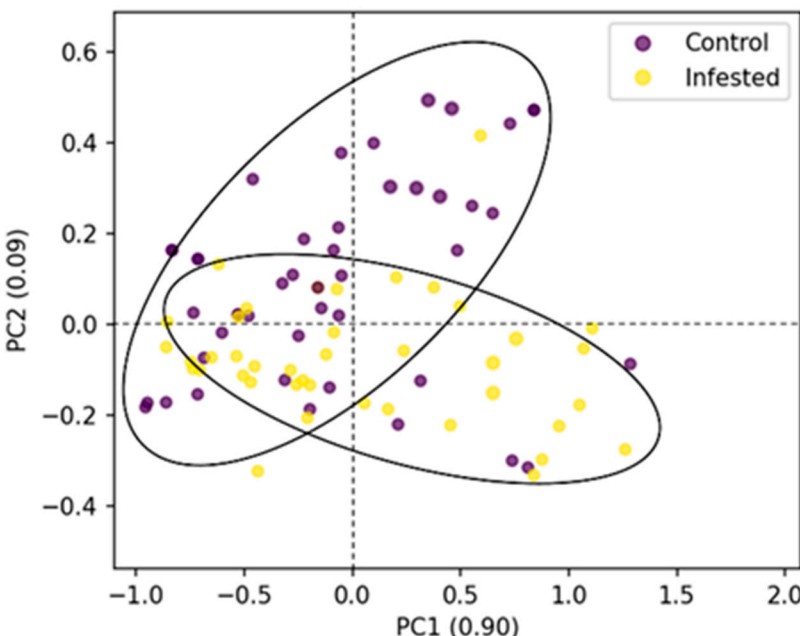

**Figure 4.** Principal component analysis of two types of apple sample tissues for the Fuji cultivar computed from the mean spectral data of the whole fruit.

### 3.2. Feature Selection of Acoustic Data

A total of 21 features including the time and frequency domains were measured from the impulse signals of the CM-infested and control apple samples. The selected features were chosen based on the correlation between the all the 21 features and the two classes of apple samples. Larger correlation values, closer to 1, indicated better correlation results. The impulse signal features of the number of zero-crossing, entropy, number of peaks, kurtosis, root mean square, and mean absolute deviation showed the highest correlation results with minor changes from one cultivar to another, such as Granny Smith, illustrating that the energy spectral density displayed high correlation. Table 2 presents the top six features with high correlation per category, which were applied in the classification step of fusion with the HSI data to classify the CM infestation in apples. The Pearson correlation coefficients of these six selected features were higher than 30% in Fuji, 58% in Gala, 60% in Granny Smith, and 45% in the combined cultivars [5]. The classification models utilized in this study for analyzing the impulse signals were applied to two datasets: one comprising the complete 21 features and the other consisting of only six selected features.

**Table 2.** The selected features applied to Fuji, Gala, Granny Smith, and all cultivars with their correlation coefficients [5].

| Fuji | Gala | Granny Smith | All Cultivars |
|---|---|---|---|
| Number of zero crossing ($r^2 = 0.68$) | Number of peaks ($r^2 = 0.88$) | Number of peaks ($r^2 = 0.79$) | Number of peaks ($r^2 = 0.79$) |
| Number of peaks ($r^2 = 0.66$) | Entropy ($r^2 = 0.82$) | Number of zero crossing ($r^2 = 0.78$) | Number of zero crossing ($r^2 = 0.76$) |
| Entropy ($r^2 = 0.45$) | Number of zero crossing ($r^2 = 0.82$) | Entropy ($r^2 = 0.78$) | Entropy ($r^2 = 0.68$) |
| Kurtosis ($r^2 = 0.40$) | Mean absolute deviation ($r^2 = 0.67$) | Mean absolute deviation ($r^2 = 0.72$) | Kurtosis ($r^2 = 0.59$) |
| Energy spectral density ($r^2 = 0.38$) | Kurtosis ($r^2 = 0.67$) | Kurtosis ($r^2 = 0.68$) | Mean absolute deviation ($r^2 = 0.58$) |
| Peak frequency ($r^2 = 0.30$) | Root mean square ($r^2 = 0.58$) | Variance ($r^2 = 0.59$) | Root mean square ($r^2 = 0.44$) |

Reprinted with permission from Elsevier [5]. 2023, Khaled et al., (2022).

### 3.3. Classification Models of the Individual Acoustic and HSI Datasets

Table 3 shows an example of a performance comparison of all the classifiers used for the classification of CM-infested Gala apples, with AdaBoost having the best performance. The results of the classification of normal and CM-infested apples using all features of the acoustic and HSI datasets are shown in Table 4. Between the two datasets, the acoustic data gave higher classification rates than the mean spectral hyperspectral data. The acoustic

data from the Gala apples were better classified using the AdaBoost ensemble learning method, achieving an accuracy of up to 97% for the test set data. The best classification accuracy for the HSI method was obtained for Fuji apples at 88% using the AdaBoost ensemble classifier. For the combination of all three cultivars, while the acoustic method achieved an acceptable classification rate in the lower 90% range, the HSI yielded a poor classification accuracy. The lower classification results from the combined samples could be attributed to the different textural and surface color characteristics such as the different pigmentation of the skin of the three apple cultivars, caused extra biological variability into the model. The pigmentation in the Granny Smith cultivar, for example, is green (non-red), while the pigmentation in the skin of the other two cultivars is red/pink [29].

**Table 3.** Comparison of the performance of different classifiers in the classification of Gala apples as units of %.

| Classifier Model | Gala–All Features (%) | | | | |
|---|---|---|---|---|---|
| | Accuracy | Standard Deviation | Precision | Recall | F1-Score |
| SVM | 64 | 11 | 64 | 65 | 60 |
| RF | 96 | 2 | 96 | 96 | 96 |
| kNN | 68 | 5 | 63 | 62 | 62 |
| DT | 96 | 2 | 96 | 97 | 97 |
| LDA | 96 | 3 | 96 | 93 | 94 |
| NB | 62 | 10 | 67 | 68 | 62 |
| Ridge | 96 | 3 | 96 | 95 | 96 |
| GB | 96 | 3 | 96 | 97 | 97 |
| QDA | 74 | 5 | 75 | 79 | 73 |
| ET | 96 | 2 | 96 | 96 | 96 |
| AB | 97 | 1 | 96 | 97 | 97 |

SVM: Support Vector Machine, RF: Random Forest, kNN: k-Nearest Neighbors, DT: Decision trees, LDA: Linear Discriminant Analysis, NB: Naïve Bayes, GB: Gradient Boosting, QDA: Quadratic Discriminant Analysis, ET: Extra Trees, AB: AdaBoost.

**Table 4.** The test-set classification results based on different sources of data from each individual sensor using the ensemble AdaBoost classifier as units of %.

| Cultivar | Features | Variables | Accuracy | Recall | Precision | F1 Score |
|---|---|---|---|---|---|---|
| Fuji | Full-HSI | 241 | 88 | 88 | 91 | 88 |
| | Acoustic | 21 | 90 | 87 | 89 | 88 |
| Gala | Full-HSI | 241 | 79 | 62 | 67 | 79 |
| | Acoustic | 21 | 97 | 97 | 96 | 97 |
| GS | Full-HSI | 241 | 71 | 71 | 71 | 71 |
| | Acoustic | 21 | 95 | 92 | 91 | 91 |
| Combined | Full-HSI | 241 | 64 | 65 | 68 | 64 |
| | Acoustic | 21 | 94 | 93 | 93 | 93 |

HSI: Hyperspectral Imaging; GS: Granny Smith. All dependent results are percentage (%) scores.

### 3.4. Classification of Each Sensing Method Dataset based on Selected Features

The results of the machine learning classification based on the HSI features extracted by PCA and the acoustic features selected by the Pearson correlation method are presented in Table 5. Overall, the PCA-based HSI models showed better performance than the models based on the full HSI spectra, whereas the dimensionality of the data was decreased significantly from 241 to 15, 10, or 5 features. This improved classification performance was due to the reduction in both the dimension of the data and the redundancy (some wavelengths) of the variable. However, for the acoustic models with the selected features a slight decrease in the classification performance was observed because the dimensions of the full-scale acoustic data were already low (21). Therefore, with the feature selection process and removal of some of the information, the accuracy was reduced, though not so

significant [5]. This slight decrease in the classification rate was compensated for by having a model with only six features in comparison to twenty-one.

**Table 5.** Selected features applied to Fuji, Gala, Granny Smith, and all cultivars [5].

| Cultivar | Features | Variables | Accuracy | Recall | Precision | F1 Score |
|---|---|---|---|---|---|---|
| Fuji | PCA-HSI | 10 | 88 | 91 | 88 | 90 |
| | Acoustic | 6 | 84 | 92 | 90 | 91 |
| Gala | PCA-HSI | 5 | 86 | 50 | 43 | 86 |
| | Acoustic | 6 | 95 | 97 | 96 | 97 |
| GS | PCA-HSI | 10 | 71 | 71 | 71 | 71 |
| | Acoustic | 6 | 93 | 91 | 89 | 90 |
| Combined | PCA-HSI | 15 | 69 | 70 | 70 | 69 |
| | Acoustic | 6 | 92 | 91 | 92 | 91 |

HSI: Hyperspectral Imaging; GS: Granny Smith. All dependent results are in percentage (%) scores.

*3.5. Classification Based on Data Fusion from Acoustic and HSI*

In the low-level data fusion application, the acoustic dataset was directly concatenated with the HSI dataset. The results of the classification of CM-infested apples for the three cultivars are presented in Table 6. In the case of the Gala cultivar, the classification performance of the low-level data fusion model was superior to each of the individual acoustic and HSI models, with all the performance metrics surpassing 98% for the test set. The combination of the acoustic and HSI improved the classification accuracy for Gala apples by 24% compared with the full-HSI spectra and by approximately 2% compared to the full acoustic dataset. Particularly important is the perfect recall result for the Fuji and Gala apple cultivars. The implication of the 100% result is that all infested apples were 100% correctly classified with zero false negatives. The high misclassification of infested GS apples, which had a clearly different color and surface reflectance, may be attributed to the skin pigmentation and reflection during the HSI scanning. This pigmentation effect was also reflected in the combined data from all the three cultivars.

**Table 6.** The performance of the classification models based on the complete data fusion as units of %.

| Cultivar | Features | Variables | Accuracy | Recall | Precision | F1 Score |
|---|---|---|---|---|---|---|
| Fuji | Acoustic–HSI | 21 + 241 | 98 | 100 | 97 | 98 |
| Gala | Acoustic–HSI | 21 + 241 | 98 | 100 | 98 | 99 |
| GS | Acoustic–HSI | 21 + 241 | 92 | 91 | 97 | 94 |
| Combined | Acoustic–HSI | 21 + 241 | 90 | 93 | 92 | 93 |

HSI: Hyperspectral Imaging; GS: Granny Smith. All dependent results are based on percentage (%) scores.

With mid-level fusion, the optimum features separately extracted by Pearson correlation and PCA for the acoustic and HSI datasets, respectively, were merged as a single matrix and then used for classification analysis (Table 7). The mid-level data fusion showed an improvement for all the three apple cultivars compared with the low-level results (Table 5). For example, the performances of the Gala apple parameters of accuracy, recall, precision, and F1-score were 98%, 98%, 100%, and 99%, respectively. These high classification rates of the mid-level data fusion compared with the low-level data were especially noticeable in the combined samples from all the cultivars. This was due to having a general model capable of classifying CM-infested apples regardless of the apple cultivar. Using the mid-level fusion approach, it was possible to classify CM-infested apples in a sample of the three different cultivars with an accuracy, recall, precision, and F1 score of 94%, 97%, 95%, and 96%, respectively.

**Table 7.** The classification performance based on the fusion of selected acoustic and HSI features as units of %.

| Cultivar | Features | Variables | Accuracy | Recall | Precision | F1 Score |
|---|---|---|---|---|---|---|
| Fuji | Acoustic–PCA-his | 6 + 6 | 94 | 97 | 94 | 96 |
| Gala | Acoustic-PCA-his | 6 + 6 | 97 | 97 | 100 | 98 |
| GS | Acoustic-PCA-his | 6 + 6 | 88 | 91 | 92 | 91 |
| Combined | Acoustic-PCA-his | 6 + 6 | 94 | 97 | 95 | 96 |

HSI: Hyperspectral Imaging; GS: Granny Smith. All dependent results are based on percentage (%) scores.

## 4. Conclusions

In this study, the fusion of acoustic and HSI sensor data obtained from apples was investigated to classify CM-infested apples. The features were fused using low- and mid-level approaches and with the application of AdaBoost, a predetermined best classifier. The performance of the classifications based on individual raw data was improved by the fusion methods leading to improved results. The results showed that the combined selected features (mid-level fusion), selected by the correlation coefficient or PCA methods, were better than using all the combined features (low-level fusion) in the classification of CM-infested apples. This improvement is particularly important in the case of the combined apples, where the data fusion gave accuracy, recall, precision, and F1 scores of 94%, 97%, 95%, and 96% in the classification of CM-infested apples regardless of the cultivar, respectively. These results prove that sensor/data fusion approach can be implemented to classify CM-infested apples and consequently help improve the sorting process for CM-damaged apples from three different cultivars.

**Author Contributions:** Conceptualization, A.A.A.; methodology, N.E. and A.Y.K.; writing—original draft preparation, N.E. and A.Y.K.; writing—review and editing, A.A.A., K.D.D. and R.T.V.; supervision, A.A.A. All authors have read and agreed to the published version of the manuscript.

**Funding:** This work was funded by the National Institute of Food and Agriculture (NIFA), the U.S. Department of Agriculture (USDA) under project award number: 2019-67021-29692, and partly by the Kentucky Agricultural Experiment Station (KAES).

**Institutional Review Board Statement:** Not relevant.

**Data Availability Statement:** Data sharing that does not conflict with the limitations of the funding agency will be made available upon request.

**Conflicts of Interest:** The authors declare no conflict of interest.

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
