# Peer review of "Classification of Codling Moth-Infested Apples Using Sensor Data Fusion of Acoustic and Hyperspectral Features Coupled with Machine Learning"

_agriculture, doi:10.3390/agriculture13040839_

Round 1
Reviewer 1 Report
This is a good paper exploring the potential use of data fusion to integrate 2 different sensory data for the non-destructive techniques to classify codling moth-infested apples.
There are several improvements that can be made on more explanation regarding the features selection on the classification method.
Please double-check your citation and list of references. References no.1 is missing from the list.

Reviewer 2 Report
The manuscript is to apply sensor data fusion of acoustic measurement and hyperspectral imaging. The topic fits to the scope of the journal.
Major comments:
- please clarify the 21 parameters used in analysis. The Fast Fourier Transform (FFT) results in a vector of coefficients. Which one was selected? Table 1 refers them in plural, suggesting more FFT coefficients were utilized. The parameters "Average number of peaks" and "Number of peaks" are unclear on the same signal. Please clarify.
- Authors state that Ada Boost was the best model based on its performance (section 3.3). Could you please insert a table listing Accuracy, Recall, Precision and F1 score for all models you have tried (on combined data)? It would be nice to see how much selected model outperformed others.
- Please improve conclusions and specify low- and mid-level fusion. Especially mid-level shall be more specific, such as correlation selected the parameters or PCA components were used in fusion.
- Please check carefully references and citations. Reference No. 1 is empty, so probably all reference numbers are wrong.
Specific comments:
- abstract should be improved, because it is not really specific and starts with long introduction. Please make it more concise and also highlight some major results. E.g. Ada Boost name does not appear in abstract.
- keywords all overlap with title. Please change, also include 'Malus domestica' referring to the latin name of apple.
- introduction L40-59 is general, please include specific information from cited literature: what kind of sensor fusion was successful to what problem, was there any limitation, etc.
- citation [17,5] shall be [5,17] in L78
- please specify common size of selected fruit (such as average and standard deviation) L92
- please use degree unit instead of letter: 45o shall be 45° in L117 (please check all text)
- the term "6.35 mm radiused" shall be "rounded with 6.35 mm radius" or something similar L154-155
- the specification of 9 V DC is enough or please add more information about the adapter/power supply L166-167
- please always identify producer the same way: Name, City, State, Counrty. L170-173
- Table 1., No. 13: please start with capital T.
- small typo in L211: discriminate analysis is discriminant analysis, like in the following rows.
- "Model." can be removed in L215
- Eq. 3 please use multiplication sign instead of letter "x"
- "over overfitting" maybe doubled, please check. L231
- introduction of PCA should be in materials and methods L231-235
- "(ranging from 0 to 1)" better to remove as correlation can range [-1,+1] in L251
- Table 2: it would be nice to report the correlation value for all selected parameters to see the strength of their relationship
- word "modeling" can be removed L265
- "ninety percent" maybe 90% is better L272
- "All dependent results are based on percentage (%) scores." footnote was provided to Table 3-6. They are percentages, not values based on percentages. Please add unit in the table or modify text/caption to specify unit of %.
- "In terms of Gala cultivar," maybe "In case of Gala cultivar," is better L301
- "performance of Gala apples’ ..." maybe "performance of Gala apple parameters ..." L319
- "This ... rates" please correct L320
- Please check parameters and values, as 4 parameters have 5 values. L336-337
- "approach can classification accuracy" please correct L339
Reviewer 3 Report
The paper brings a combination of methods as an alternative to detect codling moth infestation in apples. This approach comes from the authors' group results which found promising results using the two methodologies separated first.
L249: "A total of 21 features including time and frequency domains were measured from the 249 impulse signals of CM infested and control apple samples." - My question is about how many repetitions the authors used per fruit. I mean, how many times has an apple passed in the sensor? For me, it is not clear in the M&M section despite I Know that were used 60 fruits per cultivar.
The authors described a very well about the result obtained. However, they failed to explain the performance of classification in different cultivars. "For example, the performance of Gala apples’ accuracy, recall, precision, and 319 F1-score were 98%, 98%, 100%, and 99%, respectively. This high classification rates of the 320 mid-level data fusion compared with the low-level is especially noticeable in the com-321 bined samples from all the cultivars." why??
